# RIV: Recursive Introspection Mask Diffusion Vision Language Model

## ABSTRACT

Mask Diffusion-based Vision Language Models (MDVLMs) have achieved remarkable progress in multimodal understanding tasks. However, these models are unable to correct errors in generated tokens, meaning they lack self-correction capability. In this paper, we propose **R**ecursive **I**ntrospection Mask Diffusion **V**ision Language Model (RIV), which equips the model with self-correction ability through two novel mechanisms. The first is Introspection Training, where an Introspection Model is introduced to identify errors within generated sequences. Introspection Training enables the model to detect not only grammatical and spelling mistakes, but more importantly, logical errors. The second is Recursive Inference. Beginning with the standard unmasking step, the learned Introspection Model helps to identify errors in the output sequence and remask them. This alternating (unmask → introspection → remask) process is repeated recursively until reliable results are obtained. Experimental results on multiple benchmarks demonstrate that the proposed RIV achieves state-of-the-art performance, outperforming most existing MDVLMs. Code and models will be released as open source.

## 1 INTRODUCTION

With the rapid developments of Large Language Models (LLMs) (Radford et al., 2018; 2019; Brown et al., 2020; Touvron et al., 2023a;b; Grattafiori et al., 2024; Yang et al., 2024; Li et al., 2023b; Bi et al., 2024; DeepSeek-AI et al., 2025a;b; Yang et al., 2025a), artificial intelligence has made remarkable progress in understanding and generating human language, paving the way for more advanced multimodal systems. Vision-Language Models (VLMs) represent an important step toward Artificial General Intelligence (AGI). For a long time, autoregressive (AR) models have been the dominant approach for VLMs (Liu et al., 2023; 2024a; Li et al., 2024; OpenAI, 2024; Team et al., 2023; Bai et al., 2023; Wang et al., 2024). Recently, the emergence of masked diffusion language models (Nie et al., 2025; Zhu et al., 2025; Gong et al., 2024; Ye et al., 2025; Wu et al., 2025; Liu et al., 2025) has introduced a strong contender in the field of vision-language modeling. Models based on masked diffusion have demonstrated impressive performance on multimodal tasks (e.g., LLaDA-V, MMaDA, Dimple, LaViDa (You et al., 2025; Yang et al., 2025b; Yu et al., 2025; Li et al., 2025)). Notably, these models offer several theoretical advantages, such as parallel decoding, enhanced controllability, and the ability to leverage bidirectional attention. These strengths make masked diffusion-based models a compelling choice for advancing the capabilities of vision-language systems.

Although MDVLMs have shown tremendous potential, they also inherit certain limitations from masked diffusion models (Ou et al., 2024; Sahoo et al., 2024; Shi et al., 2024). During the denoising process, MD-VLMs gradually unmask [MASK] tokens into general tokens. However, once a token is unmasked, it cannot be modified, even if it contains errors. This issue, known in the community as a lack of self-correction capability (Wang et al., 2025), has attracted increasing attention. Recent studies have attempted to address this limitation. ReMDM (Wang et al., 2025) introduced a remasking sampler for mask diffusion models,

enabling iterative refinement. However, it is sensitive to hyperparameters and lacks robustness. Seed Diffusion (Song et al., 2025) incorporated an edit-based perturbation process during training, allowing all tokens to be re-evaluated and thereby granting the model self-correction capabilities. Generalized Interpolation Discrete Diffusion (von Rütte et al., 2025) proposed a hybrid approach that combines masking and uniform noise, unlocking the ability for the model to correct its own mistakes. While these methods can handle basic grammatical and spelling mistakes, they rely on artificially injected perturbations and are less effective at correcting intricate reasoning errors.

To equip models with stronger self-correction capability, we propose RIV, introducing innovations at both the training and inference stages. Specifically, we present a novel Introspection Training, where an Introspection Model is integrated with the Instruction Model (the model that has undergone SFT (Wei et al., 2022) and is commonly referred to as the instruction model) to identify erroneous tokens. Compared to previous methods that rely on artificially injected perturbations, the Introspection Model is trained on real errors produced by the Instruction Model, enabling it to more effectively learn to identify subtle logical errors. Additionally, we introduce an innovative Recursive Inference. The process begins with standard unmasking, followed by the Introspection Model re-examining the output sequence to identify erroneous tokens, which are then remasked. Such recursive process enables iterative self-correction. The approach is similar to a student reviewing their answers after a test, finding mistakes, and correcting them, alternating between review and correction until the final answers are accurate. We evaluate RIV on multiple benchmarks and achieve state-of-the-art performance. In summary, our technical contributions lie in the following two-fold:

- **Introspection Training:** We introduce an Introspection Model to identify errors in the outputs, using erroneous tokens generated during training as the source of training data. This approach enables the model to detect not only basic grammatical and spelling mistakes, but more importantly, subtle errors in reasoning and logic.

- **Recursive Inference:** During inference, the model alternates among unmasking, introspection, and re-masking steps, recursively refining its generated responses to support self-correction.

## 2 RELATED WORK

### 2.1 MASK DIFFUSION VLM

The recent rapid development of MDVLMs has challenged the dominance of autoregressive paradigms in multimodal understanding and has led to impressive results. Similar to vision-language models under the autoregressive paradigm (Liu et al., 2023; 2024a; Li et al., 2024), most MDVLMs adopt an architecture that includes a vision encoder, a mask diffusion large language model, and an adapter. For example, LLaDA-V (You et al., 2025) has achieved performance comparable to LLaMA3-V using this architecture. Dimple (Yu et al., 2025) introduced a confident decoding strategy that dynamically adjusts the number of tokens generated at each step, significantly reducing the number of generation iterations. MMaDA (Yang et al., 2025b) unified multimodal understanding and generation within the mask diffusion paradigm, enhancing performance through a mixed long chain-of-thought fine-tuning strategy and UniGRPO. LaViDa (Li et al., 2025) built two vision-language models based on LLaDA (Nie et al., 2025) and Dream (Ye et al., 2025), respectively, improving training efficiency through complementary masking and boosting inference efficiency with a prefix KV cache. Although these methods achieve strong performance through innovative designs, they inevitably lack self-correction capabilities.

### 2.2 SELF-CORRECTION

Currently, research on self-correction is mainly focused on masked diffusion-based large language models. ReMDM (Wang et al., 2025) built on a solid theoretical foundation and designed a novel remasking sampler

that enables the updating of previously generated tokens. Seed Diffusion (Song et al., 2025) introduced manually designed edit-based perturbations during the second stage of training, allowing all tokens to be re-evaluated and thereby achieving more robust error correction. GIDD (von Rütte et al., 2025) explored a hybrid approach that combines masking and uniform noise, unlocking the ability of models to correct errors. Although these methods have shown promising results, they do not fully take advantage of the errors encountered during training to improve the ability to recognize mistakes. In self-correction tasks, generating samples through manually injected perturbations may cause valuable logical error samples to be overshadowed by a large number of low-level error samples.

## 3 METHOD

We propose RIV, a large-scale Mask Diffusion Vision Language Model (MDVLM) that supports Self-Correction. First, in Section 3.1, we introduce the background information of MDVLMs, consisting of the modeling approach and learning objectives . Then, in Section 3.2, we describe the overall architecture of RIV. In Section 3.3, we present the training methods, with a particular focus on Introspection Training. Finally, in Section 3.4, we introduce Recursive Inference.

### 3.1 PRELIMINARY

MDVLMs consist of a forward noising process (replacing original tokens with [MASK]) and a learnable reverse denoising process (unmasking [MASK] tokens back to the original tokens) (Austin et al., 2021; Ou et al., 2024; Nie et al., 2025; Ye et al., 2025). Let $\{\mathbf{p_m}, \mathbf{x_0}\} \sim q_{\text{data}}$ represent a sample pair, where $\mathbf{p_m}$ denotes the multimodal prompt, and $\mathbf{x_0}$ represents the response containing $L$ tokens, $[x_0^1, x_0^2, \ldots, x_0^L]$. The forward process begins with clean data $\mathbf{x_0}$ and progressively replaces the tokens in $\mathbf{x_0}$ with [MASK], eventually producing a sequence $\mathbf{x_1}$ composed entirely of [MASK] tokens. Let $\mathbf{x_t}$ denote the sequence at time step $t$, where $t \in [0, 1]$. The learnable reverse process starts from $\mathbf{x_1}$ and gradually unmasks to recover the clean data $\mathbf{x_0}$. The learning objective of the model $\theta$ can be optimized using Equation 1.

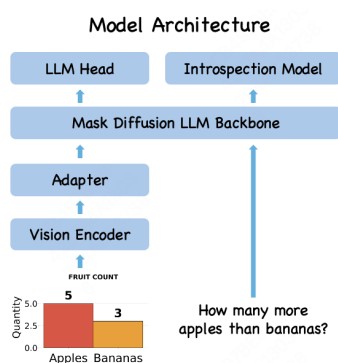

Figure 1: RIV Model Architecture. We integrated the Introspection Model into the Instruction Model to identify errors.

$$L_U(\theta) = -\mathbb{E}_{t,\mathbf{x_0},\mathbf{x_t}} w(t) \sum_{i=1}^{L} \mathbf{1}[x_t^i = [\text{MASK}]] \log p_\theta(x_0^i|\mathbf{p_m}, \mathbf{x_t}). \tag{1}$$

Here, the indicator function $\mathbf{1}[x_t^i = [\text{MASK}]]$ restricts the loss calculation to only those positions where tokens are masked. The term $w(t) \in (0, 1]$ serves as a time-dependent weighting factor.

### 3.2 MODEL ARCHITECTURE

Our proposed RIV, as illustrated in Figure 1, consists of four modules: a mask diffusion-based language model, a vision encoder, an adapter, and an Introspection Model for identifying erroneous tokens. Specifically, we use the high-performing Dream (Ye et al., 2025) as the LLM backbone. For the vision encoder, we use QwenViT (Bai et al., 2025), which supports dynamic resolution and efficiently handles visual inputs of varying sizes. The adapter is implemented as a two-layer MLP, whose primary function is to align the feature space of QwenViT with the mask diffusion paradigm. The Introspection Model is designed to identify erroneous tokens generated during the denoising process. This model consists of a transformer block and a linear layer as the output head for token classification.

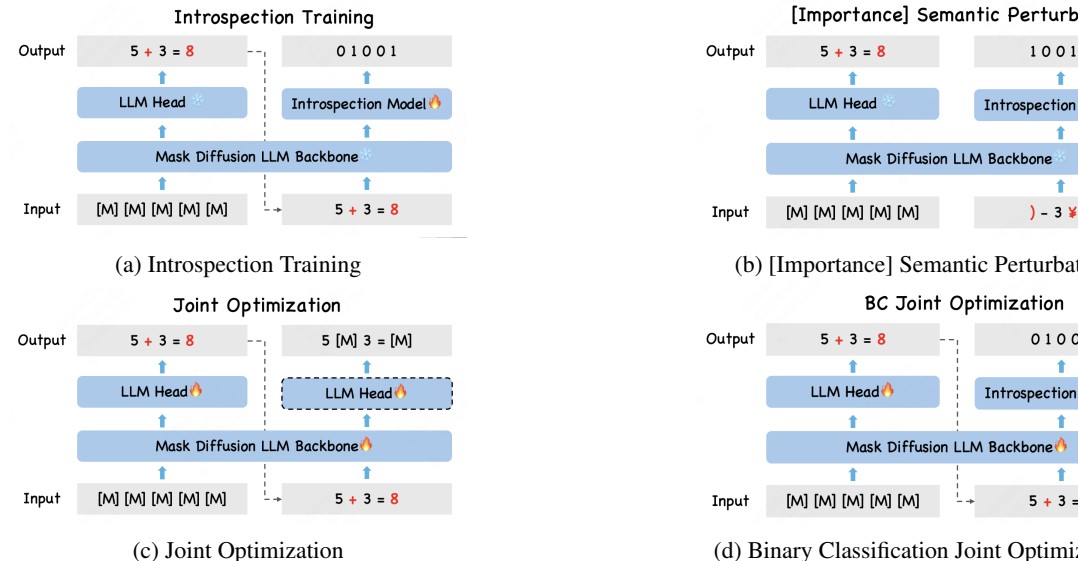

(a) Introspection Training

(b) [Importance] Semantic Perturbation

(c) Joint Optimization

(d) Binary Classification Joint Optimization

Figure 2: Comparison of Different Training Methods. The supervised ground truth in the figure is $5-3=2$. [M] denotes [MASK] token. Introspection Training is shown in 2a, where the output of the Instruction Model is used as the training data for the Introspection Model. Erroneous tokens in the output sequence are highlighted in red. The Introspection Model performs binary classification on sequences containing erroneous tokens, outputting 1 for errors and 0 otherwise. In Introspection Training, we only train the Introspection Model, while the Instruction Model is frozen. Subfigure 2b presents an ablation experiment, where sequences with random perturbations are used as inputs to the Introspection Model, and the model is responsible for identifying which tokens have been perturbed. In 2c and 2d, we jointly optimize the two objectives of unmasking and identifying erroneous tokens. In 2c, tokens that are considered erroneous are directly replaced with [MASK] in the output. In 2d, the model predicts the confidence scores of the erroneous tokens, which is consistent with subfigure 2a.

## 3.3 TRAINING STRATEGY

To equip the model with robust multimodal understanding capabilities, we carefully designed four training stages.

**Stage 1 Visual Alignment.** The objective is to align QwenViT with the feature space of the mask diffusion paradigm. In this stage, both the adapter and ViT are trained simultaneously using 4.4 million caption data.

**Stage 2 Instruction Fine-tuning.** The goal is to enhance the basic instruction-following ability of model. During this stage, we unfreeze the LLM blocks, vision encoder, and adapter. The training data consists of 10 million Mammoth (Guo et al., 2024) and 3.2 million in-house SFT data. The distribution of the in-house SFT data is shown in B.

**Stage 3 CoT Fine-tuning.** This stage aims to further enhance the reasoning ability and comprehensive multimodal understanding of model. The training parameters remain the same as Stage 2. In this stage, we use 76,000 high-quality internal chain-of-thought (CoT) (Wei et al., 2023) data samples (resampled 5 times), along with 10% of the data sampled from Stage 2. The first three stages are trained using $L_U$ (see Equation 1). For convenience, we refer to the model obtained in Stage 3 as the Instruction Model.

**Stage 4 Introspection Training.** The above three training stages are basically consistent with the conventional VLM training process. In Stage 4, as shown in the Figure 2a, we only train the Introspection Model to identify erroneous tokens. In our view, the key lies in a high-quality correction pair training set, which consists of sequences with erroneous tokens and their corresponding correct and reasonable sequences. The erroneous tokens in these correction pairs should not be entirely random but should instead be meaningful errors to some extent, as this helps the model identify complex logical errors. Specifically, during the training process, we sample $t$ from a uniform distribution $t \sim \mathcal{U}(0, 1)$ and add noise to the clean data $\mathbf{x_0}$, replacing tokens in $\mathbf{x_0}$ with [MASK] at a probability of $t$, thereby generating $\mathbf{x_t}$. Taking $\{\mathbf{p_m}, \mathbf{x_t}\}$ as the input of Instruction Model, the output is denoted as $\mathbf{x}_{\text{pred}}$. The $\mathbf{x}_{\text{pred}}$ may contain erroneous tokens. Subsequently, $\{\mathbf{p_m}, \mathbf{x}_{\text{pred}}\}$ is fed into the model, and the features from the penultimate layer of the LLM backbone are extracted as the input for the Introspection Model. The Introspection Model is required to predict whether each token in $\mathbf{x}_{\text{pred}}$ is an erroneous token. Naturally, tokens in $\mathbf{x}_{\text{pred}}$ that differ from $\mathbf{x_0}$ are treated as positive samples with ground truth $y_t^i = 1$, while identical tokens are treated as negative samples with ground truth $y_t^i = 0$ , as shown in Equation 2.

$$y_t^i = \begin{cases} 1, x_{pred}^i \neq x_0^i, \\ 0, x_{pred}^i = x_0^i. \end{cases} \qquad (2)$$

We optimize this objective using a binary cross-entropy loss function $L_I$, as described in Equation 3.

$$L_I(\theta) = -\frac{1}{L} \sum_{i=1}^{L} \left[ \log p_\theta(y_t^i | \mathbf{p_m}, \mathbf{x}_{\text{pred}}) \right] . \qquad (3)$$

This method directly leverages the erroneous outputs generated during the regular training process as training data, in contrast to artificially injected perturbations that lack semantic significance. Compared to manually designed perturbations, Introspection Training significantly improves error identification, making it possible to detect not only basic grammatical and spelling mistakes but also more complex logical errors. We conduct a comparative analysis between Introspection Training and manual perturbation training in Section 4.4.

In Stage 4, the transformer blocks in the Introspection Model are initialized using the final layer of the LLM blocks from the Instruction Model, while the output head is randomly initialized. The Instruction Model from Stage 3 remains frozen, enabling the Introspection Model to be optimized independently while preserving the abilities of the Instruction Model. This strategy prevents mutual interference between traditional unmasking learning and introspection learning, and we refer to it as Decoupled Optimization. In Section 4.4, we validate the necessity of Decoupled Optimization through ablation experiments. The training data used in this stage is the same as that used in Stage 3.

Additionally, a dynamic length training strategy is employed throughout all training stages to improve the robustness in generating responses of varying lengths. Specifically, we set a maximum response length $L_{\max}$. For each sample with an answer length $L'$, if $L' < L_{\max}$, we uniformly sample a response length $L$ from the interval $[L', L_{\max}]$, and pad the response with EOS tokens until it reaches length $L$. Further details on the training setup are provided in Section 4.1

## 3.4 RECURSIVE INFERENCE

Recursive Inference is designed to enable iterative refinement, as shown in Figure 3. Specifically, $\{\mathbf{p_m}, \mathbf{x_1}\}$ is first fed into the model, and the Instruction Model is used to perform $S$ steps of denoising until all tokens are unmasked, thereby generating $\mathbf{x}_{\text{pred}}$. Next, $\{\mathbf{p_m}, \mathbf{x}_{\text{pred}}\}$ is fed into the model, and the penultimate layer features from the LLM are used as input to the Introspection Model, which produces $\mathbf{x}_I$. Here, $\mathbf{x}_I$ represents the confidence that each token contains an error, with higher values indicating a greater likelihood of error. Tokens with confidence scores exceeding a predefined confidence threshold $c$ are replaced with [MASK].

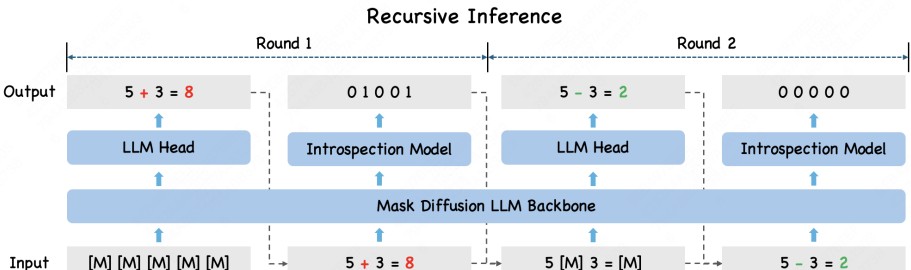

Figure 3: Recursive Inference

Let the number of erroneous tokens be denoted as $e$, and we update $S$ as $S = e//2+1$. If $e$ is zero, the model considers the sentence error-free and the inference process terminates, completing one round of inference. A maximum recursion depth $R$ is set, and this process is repeated until no errors are detected or the recursion limit is reached. The pseudocode for this procedure is provided in Appendix D, and a discussion of the time cost for Recursive Inference can be found in Appendix H.

Table 1: Benchmark Results. We compared VLMs with language model parameter sizes ranging from 7B to 8B. The AR-based models include: Qwen2-VL 7B, Qwen2.5-VL 7B (Wang et al., 2024; Bai et al., 2025), while the MD-based models include: MMaDA, Dimple, LaViDa-D, LLaDA-V (Yang et al., 2025b; Yu et al., 2025; Li et al., 2025; You et al., 2025).(-) denote results not reported.

| Model | MMMU val | MMB en-dev | MME-P | MMStar | MathVista mini | MathVerse mini-vision | AI2D | SeedB image | RealworldQA | ChartQA | DocVQA val | InfoVQA val |
|---|---|---|---|---|---|---|---|---|---|---|---|---|
| *AR Models* | | | | | | | | | | | | |
| Qwen2-VL 7B | 54.1 | - | - | 60.7 | 58.2 | - | 83.0 | - | 70.1 | 83.0 | - | - |
| Qwen2.5-VL 7B | 58.6 | - | - | 63.9 | 68.2 | 49.2 | 83.9 | - | 68.5 | 87.3 | - | - |
| *MD Models* | | | | | | | | | | | | |
| MMaDA | 30.2 | 68.5 | 1410.7 | - | - | - | - | - | - | - | - | - |
| Dimple | 45.2 | - | 1514.0 | - | 42.3 | - | - | - | - | 63.4 | - | - |
| LaViDa-D | 42.6 | 73.8 | 1463.5 | - | 42.1 | 24.1 | 69.0 | - | - | 61.0 | 56.1 | 36.2 |
| LLaDA-V | 48.6 | **82.9** | 1507.0 | **60.1** | 59.7 | 28.5 | 77.8 | **74.8** | 63.2 | 78.3 | 83.9 | 66.3 |
| **RIV** | **54.3** | 82.6 | **1647.7** | 58.3 | **60.7** | **36.2** | **80.3** | 73.1 | **65.9** | **83.9** | **89.5** | **72.3** |

## 4 EXPERIMENT

In this section, we first describe the training setup in Section 4.1. Next, Sections 4.2 and 4.3 present the evaluation setup and benchmark results, respectively. Finally, Section 4.4 reports three ablation studies, namely the impact of Self-Correction on performance, the effectiveness of Introspection Training, and the necessity of Decoupled Optimization.

### 4.1 TRAINING SETUP

Throughout all training stages, we set the maximum response length $L_{\max}$ to 512. For QwenViT (Bai et al., 2025), the token number range is set from 200 to 1337. Weight decay is set to 0, with a warmup period of 600 steps, and the learning rate schedule follows a cosine decay strategy. To reduce memory usage and improve training efficiency, we employ DeepSpeed ZeRO Stage 2. The total computational resources required for

the entire training process amount to 8,672 H800 hours. The training hyperparameters for each stage are summarized in Table 6.

## 4.2 EVALUATION SETUP

To comprehensively evaluate the effectiveness of our proposed RIV, we conduct experiments on a range of benchmarks, including multimodal reasoning and knowledge tasks such as MMMU (Yue et al., 2024), MMStar (Chen et al., 2024), MME (Fu et al., 2023), SeedBench (Li et al., 2023a), MMBench (Liu et al., 2024b), MathVerse (Zhang et al., 2024), and MathVista (Lu et al., 2023). We also test RIV on document and chart understanding tasks, including AI2D (Kembhavi et al., 2016), ChartQA (Masry et al., 2022), DocVQA (Mathew et al., 2021), and InfoVQA (Mathew et al., 2022), as well as real-world understanding tasks such as RealworldQA (x.ai, 2024). During inference, the default maximum recursion depth is set to 2, and the confidence threshold $c$ is set to 0.4. For reasoning-intensive tasks like MathVerse and MathVista, we increase the maximum recursion depth to 3. We use VLMEvalKit (Duan et al., 2025) to evaluate the model.

## 4.3 BENCHMARK RESULT

The evaluation results are presented in Table 1. As shown, RIV outperforms all mask diffusion VLM models, including LLaDA-V (You et al., 2025), Dimple (Yu et al., 2025), MMaDA (Yang et al., 2025b), and LaViDa-D (Li et al., 2025). RIV demonstrates significant advantages in document and chart understanding tasks. For instance, it achieves scores of 83.9 on ChartQA, 89.5 on DocVQA, and 72.3 on InfoVQA, substantially outperforming models like Dimple (Yu et al., 2025) and LaViDa-D (Li et al., 2025) that utilize the same language model. In reasoning-intensive tasks, RIV also exhibits strong performance, with scores of 54.3 on MMMU, 36.2 on MathVerse, and 60.7 on MathVista, surpassing other mask-based diffusion VLM models of similar scale. The Self-Correction capability plays a significant role in achieving these outstanding results. Regrettably, due to limitations in training resources and data, RIV still lags behind the advanced Qwen2.5-VL (Bai et al., 2025) in terms of performance.

## 4.4 ABLATION EXPERIMENT

**Ablation Study on Self-Correction.** We perform ablation experiments to evaluate the effect of self-correction. Specifically, we compare the performance of the Instruction Model (without self-correction) and RIV (with self-correction). As shown in Table 2, RIV consistently outperforms the Instruction Model across most benchmarks, with particularly notable improvements in reasoning-intensive tasks and certain question-answering scenarios. This demonstrates that RIV is capable of autonomously identifying and correcting logically inconsistent segments in the output sequence. These advances can be attributed to Introspection Training and Recursive Inference. For some tasks, such as MMBench (Liu et al., 2024b) and MME (Fu et al., 2023), no performance improvement is observed, as the model can directly provide answers and there is limited opportunity for further refinement. Additionally, we conduct a qualitative analysis of the unmasking, introspection, and remasking processes, which is detailed in Appendix E.

Table 2: Ablation Study on Self-Correction. SC denotes Self-Correction. (wo) indicates inference without SC, while (w) indicates inference with SC.

| Model | SC | MMMU val | MMB en-dev | MME-P | MMStar | MathVista mini | MathVerse mini-vision | AI2D | SeedB image | RealworldQA | ChartQA | DocVQA val | InfoVQA val |
|---|---|---|---|---|---|---|---|---|---|---|---|---|---|
| Instruction Model | wo | 53.8 | 82.6 | 1647.7 | 58.3 | 60.0 | 34.3 | 80.2 | 73.1 | 65.6 | 83.1 | 88.5 | 71.2 |
| **RIV** | w | **54.3** | 82.6 | 1647.7 | 58.3 | **60.7** | **36.2** | 80.3 | 73.1 | 65.9 | **83.9** | **89.5** | **72.3** |

We also investigate how the maximum recursion depth affects model performance. Specifically, we evaluate RIV on two reasoning-intensive tasks, MathVista (Lu et al., 2023) and MathVerse (Zhang et al., 2024), using maximum recursion depths of $R = \{1, 3, 6\}$. In this context, $R = 1$ corresponds to disabling the

Self-Correction, making it equivalent to using only the Instruction Model. As shown in Table 3, increasing the maximum recursion depth beyond 3 does not yield further performance gains, since the model typically produces the correct answer after two rounds of refinement.

Table 3: The Impact of Maximum Recursion Depth on Model Performance.

|  | MathVista$_{mini}$ | MathVerse$_{mini\text{-}vision}$ |
|---|---|---|
| $R = 1$ | 60.0 | 34.3 |
| $R = 3$ | 60.7 | 36.2 |
| $R = 6$ | 60.6 | 36.4 |

**Ablation Study on Introspection Training.** To demonstrate that Introspection Training is more effective than manually designe perturbations in helping the model identify subtle reasoning logic errors, we designed two sets of comparative experiments. These two experiments simulate potential reasoning errors during inference through manually designed perturbations.

*Semantic Perturbation.* In this experiment, the Introspection Model is trained without Introspection Training. Instead, random tokens are directly used to replace tokens in the input sequence, as shown in the Figure 2b.

*Importance Semantic Perturbation.* In the Semantic Perturbation Experiment, each token is replaced with an equal probability. We further consider the importance of words in the sentence. We assign importance scores to each word, with more important tokens having a higher probability of being replaced, As illustrated in the Figure 2b.

We select a model without Self-Correction capability as the baseline and train the Introspection Model on a small dataset using three different methods: Semantic Perturbation, Importance Semantic Perturbation, and Introspection Training. The three models are evaluated on the MathVista (Lu et al., 2023), with the results shown in Table 4. It can be observed that our proposed Introspection Training method achieves higher performance metrics compared to the baseline, while the other two manually injected perturbation methods show no improvement over the baseline.

Table 4: Ablation Study on Introspection Training

|  | Baseline | Semantic Perturbation | Importance Semantic Perturbation | Introspection Training |
|---|---|---|---|---|
| MathVista$_{mini}$ | 56.3 | 56.2 | 56.4 | 57.2 |

We observe the model's output results from the Semantic Perturbation Experiment and find that it could only correct basic grammatical issues and common spelling errors but failed to identify subtle logical errors. This limitation is closely related to the training strategy, as valuable logical error data is overwhelmed by a large amount of low-level error data. Although Importance Semantic Perturbation introduce an additional model to score the importance of tokens, it still fell short of capturing scenarios where the model is likely to make real errors, resulting in no significant improvement. In contrast, the Introspection Training effectively leverages the incorrect tokens generated during training, helping the model specifically learn to identify subtle logical errors. For more details, please refer to Appendix F.

**Ablation Study on Decoupled Optimization.** We validate the necessity of Decoupled Optimization through two control experiments. Using a model without Self-Correction capability as the baseline, we simultaneously optimize the two objectives of unmasking and error identification, as described below.

*Joint Optimization.* Joint Optimization does not introduce additional parameters compared to the baseline but instead directly extends the capabilities of the baseline. The training process is illustrated in the Figure 2c.

*BC(Binary Classification) Joint Optimization.* To ensure a fair comparison with Decoupled Optimization, the Introspection Model is also incorporated into the baseline, as shown in the Figure 2d. To mitigate the adverse impact of the Introspection Model's cold start on the Instruction Model, we first independently

optimize the Instruction Model using 10% of the data, and then proceed with joint training alongside the Introspection Model.

We initialize the model with parameters from the baseline and train it on a small dataset using three approaches: Joint Optimization, BC Joint Optimization, and Decoupled Optimization. In Joint Optimization and BC Joint Optimization, the LLM blocks, vision encoder, adapter, and output head are updated. In contrast, the Decoupled Optimization experiment adopted a two-step training process: first training the Instruction Model, and then independently training the Introspection Model, as shown in the Figure 2a.

Table 5: Ablation Study on Decoupled Optimization. (w) indicates inference with Self-Correction, while (wo) indicates inference without Self-Correction.

| | Baseline | Joint Optimization | BC Joint Optimization | Decoupled Optimization |
|---|---|---|---|---|
| MathVista$_{mini}$ | 58.1(wo) | 57(w) | 55.3(w) | 58.8(w) |
| | | 56.7(wo) | 55.2(wo) | 58.2(wo) |

We evaluate the three models on the MathVista (Lu et al., 2023) and find that Decoupled Optimization achieves superior performance, as shown in Table 5. Additionally, during the evaluation, we disabled the Self-Correction. The results showed that, without Self-Correction, the performance of Joint Optimization and BC Joint Optimization is worse than the baseline. We speculate that this may be due to the differing optimization spaces for the ability to identify erroneous tokens and the ability to decode tokens. Simultaneously optimizing these two objectives might introduce interference. For more details, please refer to Appendix G.

## 5 LIMITATIONS & FUTURE DISCUSSION

In this paper, we focus on equipping the model with self-correction capabilities, where the introduced Recursive Inference results in a slight increase in inference time (see H). In future work, this issue could be addressed by incorporating inference acceleration techniques specifically designed for MDMs (Wu et al., 2025; Liu et al., 2025; Ma et al., 2025; Hu et al., 2025).

## 6 CONCLUSION

We propose RIV, a Masked Diffusion-based Vision Language Model that supports self-correction. Our Introspection Training is more effective than manually designed perturbations, enabling the model to identify complex reasoning errors rather than just basic grammatical and spelling mistakes. Furthermore, our Decoupled Optimization approach allows the model to focus on error detection while preserving the performance of the instruction model. Finally, with our proposed Recursive Inference, the model fully supports self-correction. RIV achieves state-of-the-art results on multiple benchmarks, offering the research community a new perspective for exploration.

## 7 ETHICS STATEMENT

This study strictly adheres to relevant ethical guidelines and legal regulations. All data used in the study do not contain any personally identifiable information or sensitive information. As this study does not involve human subjects, ethical committee approval is not required. We acknowledge the potential risks of misuse associated with vision-language models, such as generating harmful or misleading content. To mitigate these risks, we have implemented appropriate safeguards in the model design and data processing workflows. The authors declare no conflicts of interest.

## 8 REPRODUCIBILITY STATEMENT

We place a high value on the reproducibility of our research findings. The code and model weights used in this study will be released in an open-source format.

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

## A    THE USE OF LARGE LANGUAGE MODELS

In this study, we did not use LLM. All related work is entirely based on our own algorithms.

## B    DATA

The distribution of our in-house SFT data is shown in the Figure 4.

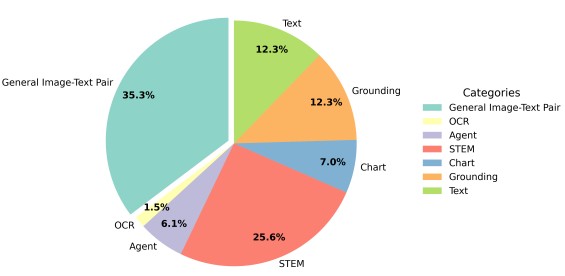

Figure 4: In-House SFT Data Distribution

## C    DETAILS OF TRAINING SETUP

The hyperparameters for all training stages can be found in Table 6.

Table 6: Training Setup of RIV

|  | Stage 1 | Stage 2 | Stage 3 | Stage 4 |
| --- | --- | --- | --- | --- |
| train param | vision encoder adapter | vision encoder adapter llm blocks | vision encoder adapter llm blocks | Introspection Model |
| data num | 4.4m | 13.2m | 1.7m | 1.7m |
| global batch size | 256 | 256 | 256 | 256 |
| max seqence length | 4096 | 5120 | 5120 | 5120 |
| adapter lr | 1e-3 | 1e-5 | 1e-6 | 0 |
| vision merger lr | 1e-6 | 2e-6 | 1e-6 | 0 |
| vision encoder lr | 1e-6 | 2e-6 | 1e-6 | 0 |
| llm blocks lr | 0 | 2e-6 | 1e-6 | 0 |
| Introspection Model lr | 0 | 0 | 0 | 1e-4 |

## D   RECURSIVE INFERENCE PSEUDOCODE

---

**Algorithm 1** Recursive Inference Pseudocode

---

Input: maximum recursion depth $R$, denoise steps $S$
Instruction Model $\theta$, Introspection Model $\theta_I$
confidence threshold $c$, response sequence $\mathbf{x}_t$
multimodal prompt $\mathbf{p}_m$
Output: $\mathbf{x}_{\text{pred}}$

1: $\mathbf{x}_t \leftarrow \{[\text{MASK}], ..., [\text{MASK}]\}$
2: **for** $r \in \{R, R-1, \ldots, 1\}$ **do**
3:    **for** $s \in \{0, 1, \ldots, S-1\}$ **do**
4:       $\mathbf{x}_t \leftarrow \theta(\mathbf{p}_m, \mathbf{x}_t)$
5:    **end for**
6:    $\mathbf{x}_{\text{pred}} \leftarrow \mathbf{x}_t$
7:    **if** $r = 1$ **then**
8:       **break**
9:    **end if**
10:   $\mathbf{x}_I \leftarrow \theta_I(\theta(\mathbf{p}_m, \mathbf{x}_{\text{pred}}))$
11:   erroneoustokens $\leftarrow \{i | x_I^i > c\}$
12:   **if** erroneoustokens $= \emptyset$ **then**
13:      **break**
14:   **end if**
15:   **for** $i \in$ erroneoustokens **do**
16:      $x_{\text{pred}}^i \leftarrow [\text{MASK}]$
17:   **end for**
18:   $\mathbf{x}_t \leftarrow \mathbf{x}_{\text{pred}}$
19:   $S \leftarrow \text{UpdateStepByErr(erroneoustokens)}$
20: **end for**

---

## E   CASE QUALITATIVE ANALYSIS

We conduct a qualitative analysis of cases from the evaluation process, as outlined below.

- Example 1. The example is taken from the 946th question in MathVista (Lu et al., 2023). Given an image 5, the task is to calculate the age difference between the two individuals in the image, with the correct answer being 16.

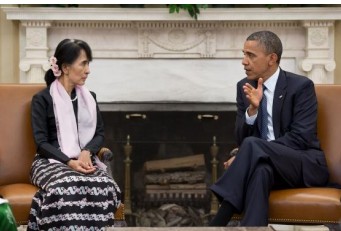

Figure 5: MathVista-946

The model performed a total of 88 inference steps. From the intermediate inference process (see 6), we can see that in step 79, the model incorrectly used the year 1965 and provided an incorrect answer of 20

years. The Introspection Model identified this error and first corrected the year to 1961. During the second check, it further noticed that $1961 - 1945 \neq 20$, prompting a second correction. Ultimately, it arrived at the correct answer of 16 years. RIV effectively optimizes its answers by reevaluating the generated results and verifying factual content, thereby producing more accurate and coherent outputs.

| Step | Model Output |
|------|--------------|
| 79 | \<think\> The person on the left is Aung San Suu Kyi, and the person on right is Barack Obama. Aung San Suu Kyi was born in 1945, and Barack Obama was born in 1961. Therefore, the age gap is 196`5` - 1945 = `20` years. \</think\>\<answer\> The answer is \$\\boxed{`20`}\$ \</answer\> |
| 80 | \<think\> The person on the left is Aung San Suu Kyi, and the person`<\|mask\|><\|mask\|>` is Barack Obama. Aung San Suu Kyi was born in 1945, and Barack Obama was born in 1961. Therefore, the age gap is 196`<\|mask\|>` - 1945 = `20` years. \</think\>\<answer\> The answer is \$\\boxed{`20`}\$ \</answer\> |
| 82 | \<think\> The person on the left is Aung San Suu Kyi, and the person on right is Barack Obama. Aung San Suu Kyi was born in 1945, and Barack Obama was born in 1961. Therefore, the age gap is 196`1` - 1945 = `20` years. \</think\>\<answer\> The answer is \$\\boxed{`20`}\$ \</answer\> |
| 83 | \<think\> The person on the left is Aung San Suu Kyi, and the person`<\|mask\|><\|mask\|>` is Barack Obama. Aung San Suu Kyi was born in 1945, and Barack Obama was born in 1961. Therefore, the age gap is 196`1` - 194`<\|mask\|>` = `<\|mask\|><\|mask\|>` years. \</think\>\<answer\> The answer is \$\\boxed{`<\|mask\|><\|mask\|>`}\$ \</answer\> |
| 87 | \<think\> The person on the left is Aung San Suu Kyi, and the person on right is Barack Obama. Aung San Suu Kyi was born in 1945, and Barack Obama was born in 1961. Therefore, the age gap is 196`1` - 1945 = `16` years. \</think\>\<answer\> The answer is \$\\boxed{`16`}\$ \</answer\> |

Figure 6: Case Qualitative Analysis 1. "Step" represents the inference step index, starting from 0, and "Model Output" shows the model's output at the corresponding inference step. Incorrect words are highlighted in red, words identified as incorrect by the Introspection Model and remasked as `<|mask|>` are shown in blue, and the corrected words are displayed in green.

- Example 2. The example is taken from the 2215th question in MathVerse (Zhang et al., 2024). Given an image 7, the task is to calculate $\cos a$ in the image, with the correct answer being $-\frac{21}{29}$.

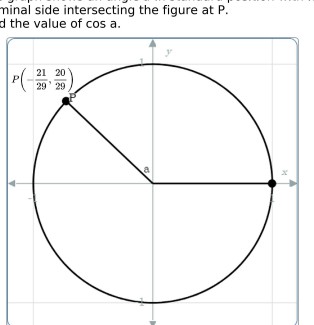

The graph shows an angle *a* in standard position with its terminal side intersecting the figure at P. Find the value of cos a.

$P\left(-\frac{21}{29}, \frac{20}{29}\right)$

Figure 7: MathVerse-2215

The model performed a total of 168 inference steps. From the intermediate inference process (see 8), we can see that in step 159, the model incorrectly added an extra "$-$" sign, which led to an incorrect result. In step 162, the "$-$" sign is correctly removed, but the answer is still not corrected at that point. In step 167, the final answer is revised to be correct.

| Step | Model Output |
|------|--------------|
| 159 | \<think\> The point P on the circumference of the unit circle has coordinates \\(( -\\frac{█21}{29}, \\frac{20}{29})\\). In the unit circle, the x-coordinate of the angle gives the cosine of the angle. Therefore, the x-coordinate of point P is \\(\\frac{21}{29}\\), which is \\(\\cos a\\). \</think\>\<answer\> The answer is $\\boxed{\\frac█21}{29}}$ \</answer\> |
| 160 | \<think\> The point P on the circumference of the unit circle has coordinates \\((\<\|mask\|\>frac\<\|mask\|\>21}{29}, \\frac{20}{29})\\). In the unit circle, the x-coordinate of a point gives the cosine of the angle. Therefore, the x-coordinate of point P is \\(\\frac{21}{29}\\), which\<\|mask\|\> \\(\\cos a\\). \</think\>\<answer\> The answer is $\\boxed{\\frac█21}{29}}$ \</answer\> |
| 162 | \<think\> The point P on the circumference of the unit circle has coordinates \\(( -\\frac█21}{29}, \\frac{20}{29})\\). In the unit circle, the x-coordinate of a point gives the cosine of the angle. Therefore, the x-coordinate of point P is \\(\\frac{21}{29}\\), which is \\(\\cos a\\). \</think\>\<answer\> The answer is $\\boxed{\\frac█21}{29}}$ \</answer\> |
| 163 | \<think\> The point P on the circumference of the unit circle has coordinates \\(( -\\frac█21}{29}, \\frac{20}{29})\\). In the unit circle, the x-coordinate of a point gives the cosine of the angle. Therefore, the x-coordinate of point P is \\(\\(\<\|mask\|\>\<\|mask\|\>21}{29}\\), which\<\|mask\|\> \\(\\cos a\\). \</think\>\<answer\> The answer is $\\boxed\<\|mask\|\>frac\<\|mask\|\>\<\|mask\|\>1}{29}}$ \</answer\> |
| 167 | \<think\> The point P on the circumference of the unit circle has coordinates \\(( -\\frac█21}{29}, \\frac{20}{29})\\). In the unit circle, the x-coordinate of a point gives the cosine of the angle. Therefore, the x-coordinate of point P is \\(\\frac{-21}{29}\\), which is \\(\\cos a\\). \</think\>\<answer\> The answer is $\\boxed{\\frac█-21}{29}}$ \</answer\> |

Figure 8: Case Qualitative Analysis 2. "Step" represents the inference step index, starting from 0, and "Model Output" shows the model's output at the corresponding inference step. Incorrect words are highlighted in red, words identified as incorrect by the Introspection Model and remasked as `<|mask|>` are shown in blue, and the corrected words are displayed in green.

- Example 3. The example is taken from the 610th question in MathVerse (Zhang et al., 2024). Given an image 9, the task is to calculate the height of the cone, with the correct answer being D.

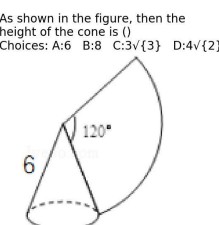

As shown in the figure, then the height of the cone is ()
Choices: A:6  B:8  C:3√{3}  D:4√{2}

120°

6

Figure 9: MathVerse-610

The model performed a total of 167 inference steps. From the intermediate reasoning process (see 10), we can see that in step 159, the model incorrectly calculated $\sqrt{32}$ as $4\sqrt{3}$. In step 160, the Introspection Model identified the error and remasked the incorrect token. In step 166, the final correct answer is provided.

| Step | Model Output |
|------|--------------|
| 159 | <think> To find the height of the cone, we can use the properties of the sector. The central angle of the sector is 120°, which is 1/3 of a full circle (360°). The radius of the sector is 6, so the arc of the sector is 1/3 * 2π * 6 = 4π. This arc becomes the circumference of the base of the cone. Let the circumference of the base be 2πr, where r is the radius of the base of the cone. So, 2πr = 4π. Solving this, we get r = 2. Now, the height of the cone is √(6² - r²) = √(36 - 4) = √(32 = 4 √ **4** . </think><answer> The answer is $\\boxed{**4**}$ </answer> |
| 160 | <think> To find the height of the cone, we can use the `<|mask|>` of the sector. The central angle of the sector is 120°, which is 1/3 of a full circle (360°). The radius of the sector is 6, so the `<|mask|><|mask|>` the sector is 1/3 * 2π * 6 = 4π. This arc becomes the circumference of the base of the cone. Let the circumference of the base be 2πr, where r is the radius of the base of the cone. `<|mask|>`, 2πr = 4π. Solving this, we get r = 2. Now, the height of the cone is √ (6² - r²) = √ (36 - 4) = √ `<|mask|><|mask|><|mask|><|mask|>` `<|mask|>` √ `<|mask|>`. </think><answer> The answer is $\\boxed{`<|mask|>`}$ </answer> |
| 166 | <think> To find the height of the cone, we can use the properties of the sector. The central angle of the sector is 120°, which is 1/3 of a full circle (360°). The radius of the sector is 6, so the arc of the sector is 1/3 * 2π * 6 = 4π. This arc becomes the circumference of the base of the cone. Let the circumference of the base be 2πr, where r is the radius of the base of the cone. So, 2πr = 4π. Solving this, we get r = 2. Now, the height of the cone is √ (6² - r²) = √ (36 - 4) = √ 32 which is 4 √ **2** . </think><answer> The answer is $\\boxed{**D**}$ </answer> |

Figure 10: Case Qualitative Analysis 3. "Step" represents the inference step index, starting from 0, and "Model Output" shows the model's output at the corresponding inference step. Incorrect words are high-lighted in red, words identified as incorrect by the Introspection Model and remasked as `<|mask|>` are shown in blue, and the corrected words are displayed in green.

# F  ABLATION STUDY ON INTROSPECTION TRAINING

- **Semantic Perturbation Experiment.** During training, semantic perturbation are injected into the model input. Specifically, for a given token $x_0^i \in \mathbf{x_0}$, we compute the cosine similarity between the embedding vector of $x_0^i$ and the embedding vectors of other tokens in the vocabulary $\mathbf{V}$. Then, we normalize these similarities using the softmax function to obtain a distribution $s(x_0^i)$, see 4.

$$s(x_0^i) = \frac{e^{cos(Embed(x_0^i), Embed(x_0^j))}}{\sum_{j=1}^{V} e^{cos(Embed(x_0^i), Embed(x_0^j))}}, x_0^j \neq x_0^i, i \in \{1, 2...L\}, j \in \{1, 2...V\}. \quad (4)$$

For each token in the sequence $\mathbf{x_0}$, we apply perturbations with a probability of $pp = 0.1$. If a token $x_0^i$ is selected for perturbation, we randomly sample a new token from the distribution $s(x_0^i)$ to replace $x_0^i$, thereby generating a correction data pair $\{\mathbf{x_0}, \mathbf{x_0'}\}$. The perturbed sequence $\mathbf{x_0'}$ is then fed into the model. The model is required to learn to identify the perturbed tokens in $\mathbf{x_0'}$. For perturbed tokens, the ground truth $y_{pp}^i$ is 1; otherwise, $y_{pp}^i$ is 0. The supervision is performed using the following loss function 5.

$$L_I'(\theta) = -\frac{1}{L} \sum_{i=1}^{L} \left[ \log p_\theta(y_{pp}^i | \mathbf{p_m}, \mathbf{x_0'}) \right]. \quad (5)$$

- **Importance Semantic Perturbation.** In Semantic Perturbation, each token has an equal probability $pp$ of being perturbed. In Importance Semantic Perturbation, we further consider the importance of words in the sentence. By utilizing KeyBERT (Grootendorst, 2020) , we pre-compute the importance score $I(x_0^i)$ for each word in the training data. We design the perturbation probability $pp(x_0^i)$ of each token as shown in Equation 6. Similar to Semantic Perturbation, if a token is selected for perturbation, a replacement token is sampled from the distribution $s(x_0^i)$. This approach ensures that semantically more important tokens

are more likely to be perturbed, and we expect the model to pay more attention to errors in key tokens. This experiment uses the same loss function as Semantic Perturbation, as shown in Equation 5.

$$pp(x_0^i) = \frac{e^{I(x_0^i)}}{\sum_{j=1}^{V} e^{I(x_0^j)}}, i \in \{1, 2...L\}, j \in \{1, 2...V\}. \tag{6}$$

## G    Ablation Study on Decoupled Optimization

- Joint Optimization. In this experiment, we still use the cross-entropy loss function $L_M(\theta)$ to optimize the objective of error identification. When $x_{\text{pred}}^i$ is the same as $x_0^i$, the ground truth $y_t^i$ is $x_0^i$; otherwise, the ground truth $y_t^i$ is [MASK]. The overall loss function can be expressed as Equation 7.

$$L(\theta) = L_U(\theta) + \alpha \cdot L_M(\theta). \tag{7}$$

  where $\alpha$ is the weight of $L_M(\theta)$, we use $\alpha = 0.5$ by default.

- BC Joint Optimization. In this experiment, we used the same loss function as Decoupled Optimization to optimize the second objective. The overall loss is also a weighted sum of $L_U(\theta)$ and $L_I(\theta)$.

## H    Time Cost of RIV

RIV performs Self-Correction in a recursive manner, but the increase in inference time is minimal. Specifically, during evaluation, we calculate the percentage increase in inference time with Self-Correction compared to without Self-Correction .

Table 7: Time Cost of RIV. The values in the table represent the percentage increase in inference time under the corresponding $R$ compared to $R = 1$ (without Self-Correction).

|         | MathVista mini | MathVerse mini-vision |
|---------|----------------|------------------------|
| $R = 3$ | 8.10%          | 10.60%                 |
| $R = 6$ | 10.30%         | 12.40%                 |

As shown in 7, when the maximum recursion depth is set to 3, the inference time on all questions in Math-Vista  (Lu et al., 2023) increased by 8.1%. When the maximum recursion depth is set to 6, the increase is only 10.3%, which remains within a manageable range.

