# OpenReview forum: "RIV: RECURSIVE INTROSPECTION MASK DIFFUSION VISION LANGUAGE MODEL"
_ICLR.cc/2026/Conference — ICLR 2026 Conference Withdrawn Submission_

### Official Review · Reviewer_VNxo · 2025-10-23

**Soundness:** 3
**Presentation:** 2
**Contribution:** 3
**Rating:** 4
**Confidence:** 3

**Summary:**

The paper proposes RIV (Recursive Introspection Mask Diffusion Vision-Language Model), which adds self-correction to masked diffusion-based VLMs. The introspection model is introduced to detect erroneous tokens. The recursive inference mechanism iteratively remasks and refines outputs based on introspection feedback.

**Strengths:**

1. Integrating the introspection model and recursive inference into base models is an original and practical direction.
2. The paper introduces and compares several training methods, and decoupled optimization, which separates the training of the introspection and instruction models, is both reasonable and experimentally justified.
3. The paper provides a detailed description of the training process across four stages. The training setup table (Appendix C) promotes reproducibility.

**Weaknesses:**

1. The paper would benefit from a more explicit comparison or discussion illustrating how RIV’s remask mechanisms differ from existing remask-based methods, such as ReMDM (Wang et al., 2025).
2. The benchmarks used in evaluation focus on accuracy metrics across multimodal QA and perception tasks. It is unclear whether the proposed model improves multi-step or causal reasoning.
3. Moreover, the effectiveness of the introspection model is only demonstrated through ablation studies rather than direct evaluation of its error-detection or correction accuracy.
4. There is no discussion of cases where the introspection mechanism fails or overcorrects.
5. The current discussion of limitations (Section 5) focuses solely on inference-time efficiency. A more comprehensive discussion would strengthen the paper’s completeness.

**Questions:**

1. Issues raised in the *Weaknesses* section.
2. How sensitive is performance to the recursion depth $R$ and confidence threshold $c$? Could adaptive stopping criteria be used to improve efficiency or stability?
3. Are there examples where the introspection model leads to degraded results (e.g., misidentifying correct tokens as errors)?

---

### Official Review · Reviewer_VEJ4 · 2025-10-25

**Soundness:** 2
**Presentation:** 2
**Contribution:** 2
**Rating:** 2
**Confidence:** 3

**Summary:**

This work proposes a self-correction strategy for mask diffusion-based vision-language models (MDVLMs). In addition to the conventional stages of MDVLM pre-training, instruction tuning, and chain-of-thought (CoT) fine-tuning, the authors introduce an Introspection Model that shares the same backbone and is trained to identify erroneous tokens in the generated sequence. During inference, the model performs Recursive Inference, where tokens predicted as incorrect are remasked and regenerated iteratively until convergence. The proposed approach aims to enable MDVLMs to detect and correct logical or semantic errors in their own outputs, thereby enhancing reasoning consistency.

**Strengths:**

The paper presents a complete training and inference framework for masked diffusion-based vision-language models, extending previous MDVLM pipelines with an introspection-based self-correction mechanism. The overall system design is coherent, and the proposed training and inference procedures are clearly defined. Experiments are conducted on a broad range of multimodal benchmarks.

**Weaknesses:**

While the paper presents Recursive Introspection as a novel mechanism for self-correction in MDVLMs, the actual methodological contribution appears incremental. The proposed introspection module essentially replaces the existing low-confidence remasking strategy (e.g., LLaDA) with a learnable classifier that predicts erroneous tokens based on intermediate features. Conceptually, this is equivalent to transforming a heuristic confidence-based rule into a trainable confidence estimator, rather than introducing a fundamentally new correction paradigm.

Moreover, the ablation analysis is incomplete. Table 2 only compares the model with and without self-correction, but does not include any quantitative comparison against prior remasking or low-confidence re-evaluation strategies. As a result, it remains unclear whether the proposed learnable introspection actually outperforms simpler or established baselines. In addition, the performance improvements reported (e.g., +0.7 on MathVista and +0.5 on MMMU) are marginal (and a large portion of tasks like MMBench, MME does not improve), raising doubt on the statistical significance of the gains.

In addition, the proposed Recursive Inference significantly increases inference cost. Each recursion round executes a full denoising process of S steps, resulting in a total computational complexity of approximately O(R \times S), where R is the maximum recursion depth. Although the authors adaptively reduce S based on the number of detected erroneous tokens, the complexity still scales linearly with R. This substantially reduces the efficiency advantage typically enjoyed by masked diffusion models.

Overall, while the idea of learning a remasking policy is reasonable, the current formulation and evidence do not convincingly demonstrate substantial novelty or empirical advantage over prior work.

**Questions:**

See Weaknesses.

**Details Of Ethics Concerns:**

N.A.

---

### Official Review · Reviewer_Zc2b · 2025-11-01

**Soundness:** 3
**Presentation:** 3
**Contribution:** 3
**Rating:** 4
**Confidence:** 4

**Summary:**

Mask Diffusion-based Vision Language Models (MDVLM) can not correct its tokens. To this end, the authors propose two core mechanisms: Introspection Training, which uses an Introspection Model trained on real errors from the Instruction Model to detect grammar, spelling, and logical errors; and Recursive Inference, which alternates unmasking, introspection, and remasking until reliable results.

Evaluated on benchmarks like MMMU, MathVista, and ChartQA, RIV outperforms most MDVLMs.

**Strengths:**

S1. This paper is well written and is easy-to-follow.

S2. The studied task is important. The motivation is also interesting.

**Weaknesses:**

W1. The paper exclusively uses the Dream LLM as the LLM backbone for RIV. It does not verify if the Introspection Training and Recursive Inference mechanisms adapt to other mainstream LLM architectures.

W2. A portion of RIV’s training relies on 3.2 million in-house SFT data, but the paper only provides a vague distribution (Figure 4) without details on data diversity, annotation quality, or potential biases. This opacity hinders reproducibility and assessment of RIV’s generalization to external, unseen data.

W3. According to the results in Tab. 1, although the proposed method is good enough to beat related MDVLMs, it still has significant gap compared with QWEN-VL families.

W4. The computational overhead of the method should be better highlighted and discussed.

**Questions:**

See above.

---

### Note · Authors · 2025-11-14

I have read and agree with the venue's withdrawal policy on behalf of myself and my co-authors.